A green garlic (Allium sativum L.) based intercropping system reduces the strain of continuous monocropping in cucumber (Cucumis sativus L.) by adjusting the micro-ecological environment of soil

Xiao Xuemei 1 2
Cheng Zhihui 2
Lv Jian 1
Xie Jianming 1
Ma Ning 1
Yu Jihua yujihua@gsau.edu.cn 1
1 College of Horticulture, Gansu Agricultural University , Lanzhou , Gansu Province , China
2 College of Horticulture, Northwest A&F University , Yangling , Shaanxi Province , China
Urban Pawel
Electronic publication date: 2019 Jul 15
Publication date: 2019
Volume: 7
Electronic Location ID: e7267
Received 2019 Mar 4; Accepted 2019 Jun 7
Copyright: ©2019 Xiao et al.
Copyright year: 2019
Copyright holder: Xiao et al.
License: This is an open access article distributed under the terms of the Creative Commons Attribution License, which permits unrestricted use, distribution, reproduction and adaptation in any medium and for any purpose provided that it is properly attributed. For attribution, the original author(s), title, publication source (PeerJ) and either DOI or URL of the article must be cited.
License URL: https://creativecommons.org/licenses/by/4.0/

Keywords: Green garlic-based cropping systems, Continuous cropping obstacle, Cucumber, Soil micro-ecological environment

Funding: Special Funds for Discipline Construction of Gansu Agricultural University GAU-XKJS-2018-220 Sheng Tongsheng Innovation Funds of Gansu Agricultural University GSAU-STS-1745 GSAU-STS-1638 National Key R&D Program of China 2018YFD0201205 This work was supported by the Special Funds for Discipline Construction of Gansu Agricultural University (GAU-XKJS-2018-220), the Sheng Tongsheng Innovation Funds of Gansu Agricultural University under grant GSAU-STS-1745 and GSAU-STS-1638, and the National Key R&D Program of China (2018YFD0201205). The funders had no role in study design, data collection and analysis, decision to publish, or preparation of the manuscript.

==============================
The continuous cropping obstacle of cucumber (Cucumis sativus L.) under facility cultivation is more prevalent in China. This is associated with an imbalance in soil microbial and ecological environment in long-term monocultures. It was postulated that intercropping with green garlic would relieve the continuous cropping obstacle of cucumber by altering the soil micro-ecology status. A pot-based experiment was conducted to investigate the green garlic-cucumber intercropping and cucumber monocropping systems. The results showed that the cucumber shoot biomass was improved by intercropping with green garlic. However, the population of soil bacteria and actinomycetes increased, while the fungal population decreased. The fatty acid methyl ester (FAME) profiles indicated that soil contained more fungal FAME biomarkers (18:1ω9c, 18:2ω6, 9) and higher fungal:bacterial ratio in the monoculture system, whereas clustering of more bacterial FAME biomarkers (cy17:0, cy19:0, 16:1ω7c10, Me16:0, 10Me17:0, 10Me18:0) was observed under intercropping conditions. Moreover, significantly (P < 0.05) higher soil invertase and alkaline phosphatase activities, organic matter, and available N, P and K contents were observed under intercropping systems. These were high in both bulk and rhizosphere soils in the intercropping system when compared to monocropping system. These findings suggest that intercropping with green garlic can alleviate continuous cropping obstacle of cucumber by improving the diverse composition of soil microbial community, enzyme activities, and nutrient availability.

Introduction

The successive plantation of the same crop on the same land results in the reduction of crop yield and quality under normal cultivation management, and this is known as the ‘continuous cropping obstacle’. Several lines of evidence suggest that increased soil-borne diseases, soil acidification, secondary salinization, and soil microbial ecology imbalance are the reasons for this phenomenon (Yao, Jiao & Wu, 2006; Zhou, Yu & Wu, 2012; Zhou et al., 2014). The problem is becoming increasingly severe in agricultural production, especially in facility cultivation, which certainly remains a crucial challenge for sustaining agriculture. Therefore, it is necessary to seek an effective alternative method to overcome this shortfall.

An intercropping system is where two or more crops are grown together in the same period and in the same field. This system not only enhances resource utilization (Hamzei & Seyyedi, 2016) but also improves the soil’s physical quality (Latif et al., 1992), eventually increasing the productivity of the crop (Wu et al., 2016). Studies have shown that the soil nutrient availability, soil enzyme activities, and microbial diversity were comparatively higher under intercropping conditions than in the monocropping system (Tian et al., 2018; Li et al., 2013a; Li et al., 2013b; Li, Lin & Zhou, 2016; Wahbi et al., 2016). Based on the benefits of intercropping system in agriculture, more and more scientists have focused on its application to relieve successive cropping obstacles. Many studies have demonstrated that the intercropping system of suitable species could alleviate the continuous cropping obstacle of the target crop, such as onion-cucumber (Zhou, Yu & Wu, 2011), cumin, anise, onion, garlic-lentil (Abdel-Monaim & Abo-Elyousr, 2012) and wheat-watermelon (Xu et al., 2015). Additionally, intercropping could reduce soil-borne diseases (Xu et al., 2015) and cause changes in soil enzyme activities as well as microbial communities (Acosta-Martínez & Cotton, 2017; Li & Wu, 2018; Dai et al., 2013).

Soil enzymes act as sensitive indicators of soil quality and play a key role in organic matter decomposition and nutrient transformation (Bowles et al., 2014). According to the previous studies, crop rotation has improved the activities of amylase, cellulase, arylsulfatase and phosphatase. Soil enzyme activities have also shown significant correlations with total organic carbon, and carbon and nitrogen microbial biomass (Balota et al., 2004). Soil microbial communities share a role in many ecological processes, such as nutrient cycling, organic matter decomposition and soil structure formation (Cotton et al., 2013). It has been observed that intercropping systems significantly reduce the fungal to bacterial ratios (F: B) and fungal biomass while increasing the soil enzyme activities (Acosta-Martínez & Cotton, 2017). A profiling study of microbial phospholipid fatty acids (PLFAs) indicated that maize-based intercropping system improved phosphate uptake by modifying the microbial communities including the dominant microbial species (He et al., 2013). Previous studies paid more attention to crop-crop and crop-vegetable intercropping systems, but till date, very few studies have been performed for the vegetable-vegetable intercropping system. Cucumber (Cucumis sativus L.) is a popular greenhouse vegetable in China, but its continuous monoculture has induced cropping obstacles. In recent years, we have tried to develop an effective method to relieve the cropping problems of cucumber. Garlic (Allium sativum L.) is effectively used in the intercropping system due to its allelopathic and antimicrobial characteristics (Khan et al., 2011; Tian et al., 2018). It has been used in several vegetable facilities to relieve the continuous cropping obstacles, such as the garlic-pepper (Khan et al., 2015), garlic-eggplant (Wang et al., 2014; Wang et al., 2015), garlic-tomato (Liu et al., 2014) and garlic-cucumber intercropping systems (Xiao et al., 2012; Xiao et al., 2013). Green garlic, similar to the Chinese chive, releases more root exudates by seeding whole garlic bulbs when compared to garlic. So, it is important to understand how and why the intercropping of green garlic influences the continuous cropping of cucumber. This study was designed to investigate the differences between green garlic-cucumber intercropping and cucumber monocropping systems.

Materials & Methods

Soil Source and Cultivation

The test soil used in this study is the dark loessial soil. The 0–30 cm layer of the soil was collected by hoe and shovel from a plastic tunnel, which had been previously planted with cucumber for five years in Lanzhou, Gansu Province, China. The chemical characteristics of the soil were as follows: electrolytic conductivity (soil/water ratio of 1:5) = 340 µS cm−1; pH value (soil/water ratio of 1:1) = 7.59; organic carbon = 14.98 g kg−1; total N = 2.15 g kg−1; available N = 110.13 mg kg−1; total P = 1.34 g kg−1; available P = 191.61 mg kg−1 and available K = 216.32 mg kg−1. The soil (17 kg) in each plastic pot (46 cm  × 40 cm in diameter and height) was mixed with 100 g of ‘GOLD FUFENG’ organic fertilizer produced by the Inner Mongolia Wofeng Agricultural Development Co., Ltd. (organic matter ≥45%, N+P2O5+K2O ≥ 10% and water content ≤ 3%), 10 g of calcium superphosphate (P2O5 ≥16.0%) and 10 g of compound fertilizer (N-P2O5-K2O:18-18-18).

The pot-based experiment included a completely randomized block design with three replicates for each of the following treatments: (1) cucumber (Cucumis sativus L. cv. Xintiandi No. 1) monocropping and (2) cucumber intercropping with green garlic (Allium sativum L. cv. G064). Ten pots were used for each treatment in one replication.

One cucumber seedling was transplanted at the centre of each pot on August 13. Twenty days later, 450 g garlic bulbs were uniformly planted around the cucumber plants at a distance of 10–12 cm from the plants. No garlic bulbs were planted for cucumber monocropping. During the experiment, the soil moisture was maintained at approximately 70% of the field’s water-holding capacity. All pots were manually irrigated and hand-weeded during crop growth.

Growth and photosynthetic pigment analyses

After an interval of 15 days, 30 days and 45 days of intercropping with green garlic, the cucumber plants were uprooted and divided into roots and shoots and were then freshly weighed (g). The dry weights were measured after drying at 75 °C for 72 h to a constant weight and after killing the enzymatic activities at 105 °C for 15 min. The fresh, fully matured young leaves (the fifth leaf below the growing point) were sampled and homogenized in 80% acetone. The absorbance of the supernatant was read at 470, 649 and 665 nm by using a spectrophotometer (UV-1800; Shimadzu, Kyoto, Japan). Then the chlorophyll (Chl a, b) and carotenoid contents were determined as described by Arnon (1949).

Soil sample collection

Soil samples were collected three times with 15 days interval after the green garlic was intercropped. Every time when the cucumber plants were dug out, the rhizosphere soil from the cucumber roots was obtained for sampling according to the method described by Zhao et al. (2016). The 0–20 cm layer of bulk soil was sampled by hand from a distance of 3–5 cm from the cucumber main root in the pot. The collected soil samples were divided into three sections. One portion of the sample was maintained at 4 °C to analyze the microbial densities, including the bacteria, actinomycetes and fungi. Another portion of the sample was stored at −70 °C for analyzing the microbial community diversity and composition. The third portion of the sample was air-dried, pulverized and passed through a 1 mm sieve for analyzing the enzyme activities, available nutrients, and pH values; or through 0.149 mm sieves for analyzing the soil organic C content.

Soil chemical properties analyses

The soil organic matter content was determined using the dilution heat K2Cr2O7 oxidation volumetric method. The pH was measured in a 1:1 soil/water suspension. The available N content was analyzed using the alkali-hydrolytic diffusion method. The available P content was extracted using 1 M sodium bicarbonate (NaHCO3) and determined by spectrophotometry. The available K content was extracted with 1 M ammonium acetate (NH4Ac) and analyzed using an atomic absorption spectrophotometer (Hitachi Z-2000, Tokyo).

Soil enzymes analyses

The activities of invertase (EC 3.2.1.26), urease (EC 3.5.1.5) and alkaline phosphatase (EC 3.1.3.1) were detected with appropriate substrates and conditions (24 h incubation at 37 °C) according to Tabatabai (1994). The activity of catalase (EC 1.11.1.6) was assayed by using the KMnO4 titration method (Johnson & Temple, 1964).

Microbial community analyses

The soil microbial densities were analyzed by using the plate dilution method according to El-Tarabily et al. (1996). The beef broth peptone medium, Gause No. 1 medium, and Martin medium were used to grow bacteria, actinomycetes and fungi, respectively. The numbers of colonies were counted, and the population per gram was calculated based on the oven-dried (105 °C) soil weight.

The structural diversity and composition of soil microbial communities were analyzed by the PLFA method. The PLFA extraction and purification procedures were performed according to the methods described by Mbuthia et al. (2015). Briefly, 6 g (freeze-dried weight) of soil was extracted using a mixture of chloroform, methanol and citrate buffer (1:2:0.8). The chloroform layer was collected and then dried under nitrogen gas at 37 °C. The neutral lipids, glycolipids and phospholipids were separated using an activated silicic acid column and eluted with 5 ml chloroform, 10 ml acetone, and 5 ml methanol, respectively. Methanol eluates were collected and then dried under nitrogen at 37 °C. After saponification and methylation, the phospholipids were extracted three times with 2 ml 1:1 (v/v) methyl tert-butyl ether/hexane solution. The fractions were identified by gas chromatography (Agilent GC-6890N; Agilent Techologies, Santa Clara, CA, USA), which was equipped with a flame ionization detector and a fused silica capillary column (25 m by 0.2 mm) using H2 as the carrier gas. The column temperature was 170 °C initially and was then increased to 250 °C at 5 °C min−1. The resulting FAMEs were analyzed by using the MIDI Sherlock microbial identification system (MIDI, Inc., Newark, DE, USA). Different fatty acids were considered to represent different microorganisms. Selected FAMEs were used as bacterial markers according to the previous studies (Kourtev, Ehrenfeld & Häggblom, 2003; Zelles, 1997; Zelles et al., 1994; Mbuthia et al., 2015), and included six Gram-positive (Gram+) bacteria (i14:0, i15:0,a15:0, i16:0, i17:0, a17:0); three Gram-negative (Gram−) bacteria (16:1ω7c, cy17:0, cy19:0) and three actinomycetes (10Me16:0,10Me17:0, 10Me18:0). The bacterial sums were calculated using the Gram+, Gram−, and actinomycete markers. According to the previous research (Frostegaård & Bååth, 1996; Frostegaård, Bååth & Tunlid, 1993), fungal indicators included two saprophytic fungal biomarkers (18:1ω9c, 18:2ω6, 9) and one mycorrhizal fungi associated biomarker (16:1ω5c). The fungal sums were calculated based on the summation of 18:1ω9c and 18:2 ω6, 9. The arbuscular mycorrhizal fungi (AMF) biomarker was excluded from fungal summation due to its unique nature in soil function as a symbiotic fungus (Olsson & Alströ, 2000). FAME 20:4ω6c was used as a marker for soil protozoa according to Stromberger, Keith & Schimdt (2012). The F:B ratio was calculated by dividing the fungal sum by the bacterial sum, and the Gram+/Gram− ratio was calculated by dividing the Gram+ sum by the Gram− sum. The individual peak data for each fatty acid was converted to molar percentages by dividing the peak area by the fatty acid molecular weight and then by dividing the total molar area of all fatty acids identified in the sample.

Statistical analyses

All data are expressed as means ± standard error. Specific differences between monocropping and intercropping treatments were determined by the least significant difference test (LSD, p < 0.05) applied after an analysis of variance (ANOVA). Box plots were graphed by using SigmaPlot 10.0. A principal component analysis (PCA) was conducted by using the CANOCO software (Version 5.0; Microcomputer Power, Ithaca, NY, USA) to determine the differences in soil microbial community structure composition between intercropping and monocropping systems for each sampling date. The soil FAME biomarkers of Gram+ bacteria, Gram− bacteria, actinomycetes, saprophytic fungi, mycorrhizal fungi, protozoa, total bacterial sums, fungal/bacterial ratio, and Gram+/ Gram− ratio were considered as indicators. The individual FAMEs transformed by square root were considered to be normally distributed dataset and decreased the coefficient of variation. Correlation analyses among the soil microbial population, enzyme activities, and chemical properties were conducted using the CORR program. The analyses were performed using SAS software (version 8.1).

Results

Growth and photosynthetic pigment analyses

As shown in Fig. 1, compared to cucumber monoculture, the shoot biomass of cucumber, when intercropped with green garlic, was significantly higher (P < 0.05) on days 45. While the higher root biomass of cucumber was observed in intercropping systems on days 30 and 45.

Figure 1 Effect of intercropping with green garlic on cucumber shoot (A) and root (B) biomass after interplanting on days 15, 30 and 45.

MC-cucumber monoculture; IC-cucumber intercropping with green garlic. Data are means ± standard error of n = 3. For each sample date, columns with same letters are not significantly different at the 5% level.

Intercropping with green garlic slightly improved chlorophyll a and total carotenoid contents of cucumber, and the effect was enhanced by extending the co-existence time (Fig. 2). Only the chlorophyll b content of cucumber was significantly increased by intercropping with green garlic on the third sampling time.

Figure 2 Effect of intercropping with green garlic on the cucumber chlorophyll a (A), chlorophyll b (B) and carotenoids (C) after interplanting on days 15, 30 and 45.

MC-cucumber monoculture; IC-cucumber intercropping with green garlic. Data are means ± standard error of n = 3. For each sample date, columns with same letters are not significantly different at the 5% level.

Soil chemical properties

The intercropping treatment significantly affected the chemical properties of both, the bulk as well as the rhizosphere soil (Table 1). In the bulk soil, the organic matter, and the available N and K contents were significantly higher (P < 0.05) under green garlic-cucumber intercropping than under cucumber monocropping at each sampling date. The available P content in the soil showed a significant increase (P < 0.05) in the intercropping treatment, except at the first sampling date. There were no significant differences in the soil pH values between the intercropping and monocropping systems. Except for pH value, other chemical properties were 20%–40% higher in the rhizosphere soil than in the bulk soil on the same sampling date. In the rhizosphere soil, the available N, P and K contents had significant increases (P < 0.05) at each sampling date, when intercropped with green garlic. Compared to monoculture, the green garlic-intercropping system showed significant improvement in the pH value on the third sampling date. These results indicate that there is a positive effect of intercropping treatment on soil chemical properties which enhanced with prolonged co-existence time.

Table 1 Soil chemical properties under intercropping and monocropping systems for different sampling dates.

Days after interplantation	Treatments	Organic matter (g kg−1)	Available N (mg kg−1)	Available P (mg kg−1)	Available K (mg kg−1)	pH value (1:1 soil: water)	
15	CB	25.96 ± 0.18 b	125.4 ± 2.9 b	180.5 ± 1.2 a	266.3 ± 1.4 b	7.51 ± 0.01 a	
	GB	27.95 ± 0.23 a	145.8 ± 2.9 a	179.4 ± 4.8 a	290.0 ± 2.5 a	7.50 ± 0.01 a	
	CR	27.92 ± 0.18 a	145.8 ± 2.9 b	192.6 ± 0.9 b	275.6 ± 2.2 b	7.44 ± 0.01 a	
	GR	28.48 ± 0.18 a	169.2 ± 2.9 a	200.9 ± 2.7 a	315.4 ± 2.6 a	7.46 ± 0.01 a	
30	CB	26.03 ± 0.16 b	107.9 ± 2.9 b	189.4 ± 4.2 b	285.1 ± 5.9 b	7.49 ± 0.02 a	
	GB	28.13 ± 0.18 a	137.1 ± 2.9 a	203.5 ± 2.7 a	427.9 ± 4.6 a	7.52 ± 0.01 a	
	CR	27.74 ± 0.23 b	131.3 ± 5.1 b	203.9 ± 0.6 b	294.4 ± 33.2 b	7.43 ± 0.03 a	
	GR	29.92 ± 0.10 a	172.1 ± 5.8 a	216.2 ± 0.7 a	431.8 ± 6.0 a	7.47 ± 0.01 a	
45	CB	25.98 ± 0.13 b	110.8 ± 5.8 b	185.8 ± 2.8 b	341.9 ± 0.6 b	7.48 ± 0.01 a	
	GB	29.28 ± 0.14 a	142.9 ± 2.9 a	214.0 ± 2.0 a	498.6 ± 2.8 a	7.51 ± 0.02 a	
	CR	28.02 ± 0.14 b	137.1 ± 2.9 b	197.2 ± 1.5 b	366.5 ± 2.0 b	7.42 ± 0.01 b	
	GR	30.30 ± 0.17 a	175.0 ± 5.1 a	223.3 ± 1.1 a	521.9 ± 2.3 a	7.47 ± 0.01 a	
Notes.

CB bulk soil from cucumber monocropping

GB bulk soil from green garlic intercropping with cucumber

CR rhizosphere soil from cucumber monocropping

GR rhizosphere soil from green garlic intercropping with cucumber

Data are means ± standard error of n = 3. Values followed by the same letter are not significantly different at the 5% level.

Soil enzyme activities

The soil enzyme activities varied widely among the cropping systems, sampling dates, and locations (Fig. 3). Intercropping with green garlic showed different effects among different enzymes. The soil catalase activity in the bulk soil of the intercropping system was significantly higher (P < 0.05) than that of the monocropping system at each sampling date. However, no significant difference in the rhizosphere soil was observed (Figs. 3A and 3B). The invertase activity in the rhizosphere soil of the intercropping system was significantly higher (15.87, 15.97, 15.72 glucose mg/g) when compared to monocropping (13.90, 11.29, 12.21 glucose mg/g) on days 15, 30 and 45 after interplanting, respectively. Invertase activity was significantly increased by 12% only on the third sampling date in the bulk soil of intercropping system (Figs. 3C and 3D). Intercropping significantly increased the urease activity by 26% in the rhizosphere soil after interplanting on day 15. However, the urease activity in the bulk soil of cucumber showed no significant effect by intercropping with green garlic (Figs. 3E and 3F). On the first sampling date, there were no significant differences in alkaline phosphatase activities among different treatments. By increasing co-existence time, the alkaline phosphatase activity in the bulk, as well as in the rhizosphere soils, was higher under intercropping than under monocropping (Figs. 3G and 3H).

Figure 3 Effect of intercropping with green garlic on the activities of soil catalase (A–B), invertase (C–D), urease (E–F) and alkaline phosphatase (G–H) in the bulk and rhizosphere soils of cucumber.

Samples were collected after interplanting on days 15, 30 and 45. CB-bulk soil from cucumber monocropping; GB-bulk soil from green garlic intercropping with cucumber; CR-rhizosphere soil from cucumber monocropping; and GR-rhizosphere soil from green garlic intercropping with cucumber. Data are means ± standard error of n = 3. For each sample date, lines with same letters are not significantly different at the 5% level.

Culturable soil microorganisms

Changes in soil microbial densities were shown in Fig. 4. The number of culturable bacteria in the bulk soil and actinomycetes in both the bulk and the rhizosphere soils of the intercropping system were significantly higher than those of the monocropping system at each sampling date (Figs. 4A and 4B and 4E and 4F). In contrast, intercropping significantly decreased the densities of culturable fungi by 27%, 67%, and 66% in the bulk soil and 41%, 35%, and 61% in the rhizosphere soil than that under the monocropping system on days 15, 30, and 45 after interplanting, respectively (Figs. 4C and 4D). Obviously, the differences in the culturable fungi between the intercropping and monocropping systems increased with extended co-existence time.

Figure 4 Effect of intercropping with green garlic on the number of bacteria (A–B), fungi (C–D) and actinomycetes (E–F) in the bulk and rhizosphere soils of cucumber.

Samples were collected after interplanting on days 15, 30 and 45. CB-bulk soil from cucumber monocropping; GB-bulk soil from green garlic intercropping with cucumber; CR-rhizosphere soil from cucumber monocropping; and GR-rhizosphere soil from green garlic intercropping with cucumber. Data are means ± standard error of n = 3. For each sample date, lines with same letters are not significantly different at the 5% level.

Microbial community diversity and composition

The total amount of PLFAs, which is an indicator of the total microbial biomass, was represented by total FAMEs as shown in Fig. 5. Intercropping with green garlic significantly increased (P < 0.05) the total microbial biomass in both the bulk and the rhizosphere soils compared to that of cucumber monoculture. The total microbial biomass in the monocropping system was constantly decreased with increasing co-existence time. Thirty PLFAs with chain lengths from C11 to C20 were identified in all the soil samples. However, only 16 PLFAs that were used as biomarkers of specific microbial groups were found in the soils; these are listed in Tables 2A and 2B. The sum of Gram+ and actinomycete bacterial FAMEs were significantly higher under green garlic/cucumber intercropping than under cucumber monocropping in both bulk and rhizosphere soils at all sampling dates. After interplanting, the sum of Gram− bacterial FAMEs showed a significant increase by intercropping with green garlic in both the bulk and rhizosphere soils on days 45. The marker for AMF (16:1ω5c) was significantly higher under intercropping than under monocropping system. However, the sum of saprophytic fungal FAMEs was lower under intercropping than under monocropping. Moreover, compared to monocropping, intercropping treatment significantly increased the protozoal FAME (20:4ω6c) in both the bulk and rhizosphere soils.

Figure 5 Total fatty acid methyl esters (FAMEs), (moles%) in the bulk (A) and rhizosphere (B) soils of cucumber under intercropping and monocropping systems on different sampling dates.

Samples were collected after interplanting on days 15, 30 and 45. CB-bulk soil from cucumber monocropping; GB-bulk soil from green garlic intercropping with cucumber; CR-rhizosphere soil from cucumber monocropping; and GR-rhizosphere soil from green garlic intercropping with cucumber. Data are means ± standard error of n = 3. For each sample date, lines with same letters are not significantly different at the 5% level.

Table 2 Fatty acid (mol%) composition in the rhizosphere and bulk soils under intercropping and monocropping systems for different sampling dates.

Data are means ± standard error of n = 3. Values followed by the same letter are not significantly different at the 5% level.

Fatty acids	15 days after interplantation	30 days after interplantation	45 days after interplantation	
	Monocropping	Intercropping	Monocropping	Intercropping	Monocropping	Intercropping	
(A) Fatty acid (mol%) composition in the rhizosphere soils under intercropping and monocropping systems for different sampling dates.	
G+							
i14:0	0.80 ± 0.03 a	0.71 ± 0.01 b	0.54 ± 0.02 a	0.57 ± 0.02 a	0.60 ± 0.03 a	0.52 ± 0.02 a	
i15:0	6.95 ± 0.07 a	6.25 ± 0.14 b	5.82 ± 0.14 a	6.21 ± 0.12 a	5.98 ± 0.05 b	6.37 ± 0.13 a	
a15:0	3.93 ± 0.09 a	4.06 ± 0.21 a	3.34 ± 0.04 a	3.05 ± 0.06 b	3.39 ± 0.18 a	3.75 ± 0.07 a	
i16:0	3.32 ± 0.02 a	3.57 ± 0.20 a	2.97 ± 0.08 a	2.57 ± 0.06 b	2.39 ± 0.09 b	2.83 ± 0.08 a	
i17:0	2.74 ± 0.17 b	4.21 ± 0.08 a	2.94 ± 0.09 a	3.32 ± 0.19 a	3.01 ± 0.05 b	3.69 ± 0.08 a	
a17:0	2.60 ± 0.19 b	3.58 ± 0.17 a	2.87 ± 0.04 b	3.88 ± 0.09 a	2.42 ± 0.16 b	3.75 ± 0.07 a	
Sum(G+)	20.34 ± 0.33 b	22.38 ± 0.29 a	18.49 ± 0.11 b	19.59 ± 0.06 a	17.80 ± 0.19 b	20.91 ± 0.20 a	
G−							
16:1ω7c	11.11 ± 0.15 a	8.92 ± 0.15 b	11.60 ± 0.10 a	11.49 ± 0.15 a	9.48 ± 0.09 b	11.55 ± 0.14 a	
cy17:0	2.41 ± 0.10 a	2.48 ± 0.21 a	2.35 ± 0.05 a	2.55 ± 0.13 a	2.37 ± 0.09 b	3.06 ± 0.14 a	
cy19:0	5.35 ± 0.21 a	5.30 ± 0.10 a	5.03 ± 0.04 a	5.01 ± 0.17 a	4.66 ± 0.13 b	5.45 ± 0.20 a	
Sum(G−)	18.87 ± 0.38 a	16.70 ± 0.04 b	18.99 ± 0.09 a	19.05 ± 0.15 a	16.51 ± 0.25 b	20.05 ± 0.38 a	
Actinomycetes							
10Me 16:0	14.04 ± 0.13 a	13.15 ± 0.20 b	12.06 ± 0.15 b	13.98 ± 0.33 a	15.36 ± 0.18 b	20.61 ± 0.12 a	
10Me 17:0	0.77 ± 0.03 a	0.66 ± 0.02 b	0.93 ± 0.01 a	0.56 ± 0.01 b	0.48 ± 0.03 a	0.60 ± 0.05 a	
10Me 18:0	4.57 ± 0.16 b	7.18 ± 0.28 a	3.14 ± 0.03 b	4.72 ± 0.11 a	7.71 ± 0.06 a	7.40 ± 0.12 a	
Sum(A)	19.38 ± 0.09 b	20.99 ± 0.31 a	16.13 ± 0.17 b	19.26 ± 0.39 a	23.55 ± 0.10 b	28.62 ± 0.10 a	
Fungi							
18:1ω9c	7.30 ± 0.11 a	6.47 ± 0.10 b	7.09 ± 0.09 a	5.80 ± 0.09 b	6.46 ± 0.12 a	4.88 ± 0.14 b	
18:2ω6, 9	3.05 ± 0.08 a	3.27 ± 0.18 a	3.28 ± 0.03 a	1.69 ± 0.08 b	2.70 ± 0.09 a	1.24 ± 0.06 b	
Sum(F)	10.35 ± 0.19 a	9.74 ± 0.27 a	10.37 ± 0.06 a	7.49 ± 0.09 b	9.16 ± 0.03 a	6.12 ± 0.09 b	
AMF							
16:1ω5c	3.33 ± 0.09 b	4.70 ± 0.16 a	2.34 ± 0.04 b	2.54 ± 0.03 a	2.41 ± 0.03 b	3.32 ± 0.08 a	
Protozoa							
20:4ω6c	1.11 ± 0.05 b	1.75 ± 0.06 a	0.43 ± 0.03 b	0.69 ± 0.01 a	0.46 ± 0.02 b	0.66 ± 0.01 a	
(B) Fatty acid (mol%) composition in the bulk soils under intercropping and monocropping systems for different sampling dates.	
G+							
i14:0	0.62 ± 0.02 a	0.63 ± 0.03 a	0.57 ± 0.02 a	0.66 ± 0.02 a	0.58 ± 0.01 a	0.54 ± 0.03 a	
i15:0	6.20 ± 0.00 a	6.06 ± 0.09 a	5.32 ± 0.06 b	6.64 ± 0.08 a	5.46 ± 0.17 a	5.89 ± 0.14 a	
a15:0	3.45 ± 0.09 a	3.37 ± 0.04 a	3.24 ± 0.03 b	3.82 ± 0.08 a	3.34 ± 0.10 a	3.55 ± 0.08 a	
i16:0	3.02 ± 0.03 b	2.83 ± 0.06 a	2.68 ± 0.03 a	3.02 ± 0.16 a	2.82 ± 0.08 a	3.07 ± 0.11 a	
i17:0	2.12 ± 0.02 b	4.25 ± 0.07 a	3.03 ± 0.03 a	3.02 ± 0.07 a	1.98 ± 0.03 b	3.86 ± 0.15 a	
a17:0	2.12 ± 0.03 b	3.55 ± 0.03 a	2.76 ± 0.09 a	2.99 ± 0.02 a	2.01 ± 0.01 b	3.14 ± 0.10 a	
Sum(G+)	17.53 ± 0.08 b	20.69 ± 0.20 a	17.61 ± 0.16 b	20.14 ± 0.28 a	16.19 ± 0.09 b	20.04 ± 0.34 a	
G−							
16:1ω7c	8.25 ± 0.13 a	8.53 ± 0.13 a	11.11 ± 0.01 a	9.54 ± 0.16 b	8.83 ± 0.22 b	10.51 ± 0.15 a	
cy17:0	1.88 ± 0.04 b	2.58 ± 0.08 a	2.41 ± 0.10 a	2.38 ± 0.07 a	1.94 ± 0.01 b	2.64 ± 0.03 a	
cy19:0	4.98 ± 0.04 a	4.72 ± 0.01 b	4.67 ± 0.02 b	6.03 ± 0.06 a	4.47 ± 0.10 a	4.52 ± 0.23 a	
Sum(G−)	15.11 ± 0.13 b	15.83 ± 0.20 a	18.19 ± 0.11 a	17.96 ± 0.15 a	15.24 ± 0.14 b	17.68 ± 0.11 a	
Actinomycetes							
10Me 16:0	13.94 ± 0.16 b	16.53 ± 0.27 a	12.77 ± 0.08 b	16.46 ± 0.19 a	14.22 ± 0.23 b	18.24 ± 0.12 a	
10Me 17:0	0.59 ± 0.01 a	0.51 ± 0.01 b	0.58 ± 0.03 a	0.70 ± 0.05 a	0.71 ± 0.04 a	0.51 ± 0.01 b	
10Me 18:0	3.97 ± 0.03 a	3.15 ± 0.13 b	2.44 ± 0.08 b	4.59 ± 0.17 a	5.72 ± 0.06 b	6.52 ± 0.14 a	
Sum(A)	18.50 ± 0.16 b	20.19 ± 0.34 a	15.79 ± 0.11 b	21.75 ± 0.07 a	20.66 ± 0.14 b	25.27 ± 0.09 a	
Fungi							
18:1ω9c	7.75 ± 0.02 a	7.31 ± 0.07 b	6.61 ± 0.22 a	7.04 ± 0.14 a	7.06 ± 0.04 a	4.64 ± 0.11 b	
18:2ω6, 9	4.25 ± 0.15 a	1.76 ± 0.03 b	4.52 ± 0.16 a	2.40 ± 0.11 b	1.86 ± 0.04 b	2.48 ± 0.16 a	
Sum(F)	12.00 ± 0.14 a	9.07 ± 0.08 b	11.13 ± 0.06 a	9.45 ± 0.06 b	8.91 ± 0.02 a	7.12 ± 0.06 b	
AMF							
16:1ω5c	2.70 ± 0.03 b	2.80 ± 0.01 a	2.31 ± 0.02 b	2.95 ± 0.08 a	2.51 ± 0.03 b	3.48 ± 0.05 b	
Protozoa							
20:4ω6c	0.40 ± 0.01 b	0.47 ± 0.01 a	0.33 ± 0.02 b	0.57 ± 0.00 a	0.37 ± 0.02 b	0.59 ± 0.01 a	
Notes.

Data are means ± standard error of n = 3. Values followed by the same letter are not significantly different at the 5% level.

The relative abundance of indicator FAMEs was shown in Fig. 6. Intercropping with green garlic significantly improved the accumulation of Gram+ bacteria, mycorrhizal fungi, and total bacteria in bulk and rhizosphere soils. Interestingly, the relative abundance of Gram− bacteria showed little difference between the monocropping and intercropping systems. In contrast, biomarkers of saprophytic fungal and F:B ratio were more gathered in cucumber monoculture (both bulk and rhizosphere soils).

PCA based on PLFA data showed that intercropping with green garlic distinctly affected the soil microbial community composition when compared to cucumber monoculture (Fig. 7). On three different sampling dates, the PC1 of the FAMEs biomarkers accounted for 83.5%, 78.6% and 90.6% of the variance, and the PC2 accounted for 9.8%, 18.8%, and 6.0% of the variance, respectively. Bulk and rhizosphere soils in cucumber monoculture system showed much greater relative proportions of saprophytic fungal and higher F:B ratio on each sampling date (Figs. 7A–7C). However, the intercropping system showed more differences in different sampling dates between bulk and rhizosphere soils. After interplanting, more total bacteria, AMF and protozoan indicators were found in the rhizosphere soil on days 15. Moreover, the Gram+, Gram− actinomycetes, total bacteria, and protozoa FAMEs were clustered together in rhizosphere soil, while greater relative proportions of AMF and Gram+: Gram− were found in the bulk soil on days 45 after interplanting. To better understand the relationship between the changes in community composition under different cropping systems and co-existence time, the FAME profiles for each system were analyzed independently on different sampling dates (Figs. 7D and 7E). The analysis showed a total fungal FAMEs cluster in the monocropping system, while a total bacterial FAMEs cluster in the intercropping system. Additionally, the effect was increasingly more distinct with increasing co-existence time.

Figure 6 Box plots of relative abundance of indicator FAMEs in the rhizosphere and bulk soils of cucumber under intercropping and monocropping systems.

Each group is composed of one or more FAMEs and is associated with particular taxon (see Methods). The horizontal line in the box was the median, and the upper and lower “hinges” are the first and third quartiles, respectively. The upper and lower “whiskers” extend to the highest or the lowest value within 1.5 times the inter-quartile range (the distance between the first and third quartiles). A, B, C, D, E, F, G, H and I represent indicator Gram-positive bacteria, saprophytic fungi, total bacterial sums, Gram-negative bacteria, mycorrhizal fungi, fungal/bacterial ratio, actinomycetes, protozoa and Gram+/ Gram-ratio. CB-bulk soil from cucumber monocropping; GB-bulk soil from green garlic intercropping with cucumber; CR-rhizosphere soil from cucumber monocropping; and GR-rhizosphere soil from green garlic intercropping with cucumber. Data are means ± standard error of n = 9 (three sample dates of three replications for every sample date). For each sample date, boxes with same letters are not significantly different at the 5% level.

Figure 7 Microbial community structure according to FAMEs biomarkers (G+, G−, actinomycetes, protozoa, AMF, total bacteria, total fungi, fungal: bacterial and G+: G−) comparing different treatments during each sampling date (A, B and C) and comparing different co-existence times for monocropping and intercropping systems (D and E) .

The bacterial biomarkers identified included six Gram positive (G+) bacteria (i14:0, i15:0, a15:0, i16:0, i17:0, a17:0); three Gram negative (G−) bacteria (cy17:0, cy19:0, 16:1ω 7c) and three actinomycetes (10Me16:0, 10Me17:0,10Me18:0). The fungal markers included two saprophytic biomarkers (18:1ω 9c, 18:2ω 6, 9) and an arbuscular mycorrhizal fungi (AMF) associated biomarker (16:1 ω 5c). The total fungal sum was calculated based on the summation of 18:1ω 9c and 18:2ω 6, 9. The protozoan marker was used as 20:4ω 6c. CB-bulk soil from cucumber monocropping; GB-bulk soil from green garlic intercropping with cucumber; CR-rhizosphere soil from cucumber monocropping; GR-rhizosphere soil from green garlic intercropping with cucumber; and first, second and third sampling after interplanting on days 15, 30 and 45. Data are means ± standard error of n = 3.

Correlation analysis

The correlations among the soil chemical, microbiological, and biochemical parameters were analyzed in the monocropping (Table 3A) and intercropping systems (Table 3B). In the cucumber monoculture system, the activities of soil urease, invertase and alkaline phosphatase were positively correlated with microorganism quantities (bacteria and fungi), total microbial biomass, soil organic matter (SOM), pH value and available nutrient (N, P and K) contents. The catalase activity was significantly correlated with only the total microbial biomass. Except for catalase, other enzyme activities were all significantly (P < 0.05) or markedly significantly (P < 0.01) related to each other. The number of fungi was positively correlated with that of actinomycetes (r = 0.90, P < 0.05). The correlation between the soil chemical, microbiological, and biochemical parameters was weakened when cucumber was intercropped with green garlic (Table 3B). The soil urease activity was significantly correlated with total microbial biomass (r = 0.87, P < 0.05), available N contents (r = 0.95, P < 0.01) and SOM contents (r = 0.84, P < 0.05). The soil invertase activity was only significantly correlated with available N contents (r = 0.88, P < 0.05). The alkaline phosphatase activity was significantly correlated with the number of bacteria (r = 0.93, P < 0.01) and actinomycetes (r = 0.92, P < 0.01) as well as the available K contents (r = 0.90, P < 0.05). The urease activity was significantly correlated with invertase activity (r = 0.87, P < 0.05). However, the number of fungi was negatively correlated with the number of bacteria (r =  − 0.81, P < 0.05) and actinomycetes (r =  − 0.84, P < 0.05).

Table 3 Correlations (r) between soil chemical, microbiological, and biochemical parameters d for monocropping and intercropping systems.

	Bacteria	Actinomycetes	Fungi	Total microbial biomass	Available N	Available P	Available K	pH	Organic carbon	Urease	Invertase	Alkaline phosphatase	Catalase	
(A) Correlations (r) between soil chemical, microbiological, and biochemical parameters d for monocropping systems.	
Bacteria	–	NS	NS	0.89*	0.83*	0.93**	NS	0.91*	0.97**	0.98***	0.88*	0.97**	NS	
Actinomycetes		–	0.90*	NS	NS	0.81*	0.91*	NS	NS	NS	NS	NS	NS	
Fungi			–	NS	NS	0.89*	0.86*	NS	NS	NS	0.90*	0.83*	NS	
Total microbial biomass				–	0.91*	0.95*	NS	NS	0.92*	0.90*	0.96**	0.88*	0.83*	
Available N					–	0.92*	0.94**	0.87*	0.94**	0.82*	0.95**	0.89*	NS	
Available P						–	0.88*	0.83*	0.95**	0.92*	0.97**	0.96**	NS	
Available K							–	NS	0.87*	NS	0.88*	0.87*	NS	
pH								–	0.93**	0.92**	0.85*	0.90*	NS	
Organic carbon									–	0.93**	0.93**	0.98**	NS	
Urease										–	0.90*	0.92**	NS	
Invertase											–	0.90*	NS	
Alkaline phosphatase												–	NS	
Catalase													–	
(B) Correlations (r) between soil chemical, microbiological, and biochemical parameters for intercropping systems.	
Bacteria	–	−0.83*	−0.81*	NS	NS	NS	NS	NS	NS	NS	NS	0.93**	NS	
Actinomycetes		–	NS	NS	NS	0.88*	0.94**	NS	NS	NS	NS	0.92**	NS	
Fungi			–	NS	NS	NS	NS	NS	NS	NS	NS	NS	NS	
Total microbial biomass				–	0.84*	NS	NS	NS	NS	0.87*	NS	NS	NS	
Available N					–	0.89*	0.82*	NS	0.84*	0.95**	0.88*	NS	NS	
Available P						–	0.97**	0.82*	0.84*	NS	NS	NS	NS	
Available K							–	0.86*	0.83*	NS	NS	0.89*	NS	
pH								–	0.92**	NS	NS	NS	NS	
Organic carbon									–	0.84*	NS	NS	NS	
Urease										–	0.87*	NS	NS	
Invertase											–	NS	NS	
Alkaline phosphatase												–	NS	
Catalase													–	
Notes.

NS, Correlation is not significant

* Correlation is significant at 0.05 level.

** Correlation is significant at 0.01 level.

*** Correlation is significant at 0.001 level.

Discussion

Effect of intercropping on soil microbial structure and diversity

Soil quality depends on total microbial abundance. The previous studies have verified that the abundance of soil bacteria first decreases and then increases, whereas the abundance of fungi increases over years of continuous cropping (Fu et al., 2017; Liang et al., 2012). However, soil microbial abundance can be altered by land management practice and cropping systems. According to a few studies, compared to monocropping, intercropping modified the soil microbial community structure more effectively (Li & Wu, 2018; Li et al., 2010; Li et al., 2016). In the present study, intercropping with green garlic significantly decreased the fungal population, while increased the bacterial and actinomycetes population; similar changes were also found in the rice/watermelon intercropping soil (Ren et al., 2008). However, Zhou, Yu & Wu (2011) reported that intercropping cucumber with onion or garlic promoted bacterial as well as fungal communities. These differences might be attributed to different allelopathic and antifungal effects between green garlic and garlic. Garlic-cucumber intercropping system also done in our previous study of plastic tunnel (Xiao et al., 2012; Xiao et al., 2013), found that intercropping with garlic stimulated number of soil bacteria and actinomyces, while inhibited number of fungi, and improved soil enzymes activities. The result was similar to green garlic intercropping with cucumber in the present study.

Specific microbial population was altered according to the profiling of microbial FAMEs biomarkers in the present study, meaning that the directional change of soil microbes occurs due to intercropping with green garlic. The sum of Gram− bacteria and AMF population were significantly higher under the intercropping than monocropping system, consequently inducing the relief of continuous cropping obstacle of cucumber. Evidence suggest that Gram− bacteria could effectively degrade the phenolic compounds, which act as the main autotoxins that limit the growth of cucumber in continuous monoculture (Zhang, Pan & Li, 2010; Zhou, Yu & Wu, 2012; Zhou et al., 2018). In addition, the AMF population is used as an indicator of soil function and quality in sustainable agricultural systems (Moeskops et al., 2010; Bowles et al., 2014). The 16:1ω5c FAME marker for AMF (Olsson & Alströ, 2000) was increased when intercropped with green garlic, indicating that the intercropping system offers a more favorable soil environment for AMF population than does the continuous cucumber monocropping system.

Microbial community composition and structure is also one of the indicators of soil function and quality. Long-term continuous cropping resulted in simplification and imbalance of the microbial structure, which is a critical factor contributing to continuous cropping obstacle (Fu et al., 2017). Many researches suggested that rotational cropping could relieve cropping obstacle by shifting the rhizosphere microbial community (Acosta-Martínez & Cotton, 2017; He et al., 2019). In this study, PCA analysis showed that more bacterial FAMEs biomarkers were accumulated in the intercropped soils (GB and GR), while more fungal FAMEs biomarkers were accumulated in the monoculture soils (CB and CR). This may be due to the root interactions in the intercropping system and the root exudates released by different plant species. Many studies have confirmed that different organic acid compositions of released root exudates could affect the composition of microbial communities (Xu et al., 2015; Zhou et al., 2018). The reduction of total fungi and F:B ratio in the intercropping system might be associated with the inhibitory effect of garlic root exudates on soil-borne diseases (Khan et al., 2011).

Effect of intercropping on soil enzymes activities

Except for soil microorganisms, soil enzymes are directly involved in many biochemical processes such as cycling of nutrients, fertilizer use efficiency, etc that are essential for soil quality maintenance (Li, Lin & Zhou, 2016; Li et al., 2018). Soil invertase, urease and phosphatases are important hydrolases that play key biochemical roles in the overall process of material and energy conversion in the soil ecosystem (Gu, Wang & Kong, 2009). Soil invertase hydrolyzes sucrose to glucose and fructose, urease catalyzes the hydrolysis of urea, and phosphatase catalyzes the hydrolysis of phosphoric acid to esters and anhydrides (Saha et al., 2008). Previous studies verified that the soil enzyme activities were linked to soil microbial biomass, organic matter content, and nutrient supply (Makoi1 & Ndakidemi, 2008). Acosta-Martínez et al. (2014) demonstrated that the soil enzyme activities were positively correlated with microbial biomass and protozoan population. Similar results were obtained in our experiment. In addition, greater soil catalase, invertase, urease and alkaline phosphatase activities were exhibited under the green garlic-cucumber intercropping system, which was in accordance with other studies conducted on garlic-based intercropping systems (Xiao et al., 2012; Wang et al., 2014; Liu et al., 2014; Khan et al., 2015).

Effect of intercropping on soil nutrient

Different tillage and cropping systems can transform the soil fertility status, thereby affecting crop yield and quality. Legume-based intercropping, owing to its biological fixation of N, was considered as a popular practice in traditional agriculture. Furthermore, the availability and mobilization of P, Fe and Zn in the soil were enhanced (Li et al., 2013a; Li et al., 2013b; Scalise et al., 2015). Similarly, our study demonstrated that intercropping with green garlic led to higher N, P and K availability. Numerous studies have suggested that the modification of soil nutrient availability under intercropping conditions affects direct root-mediated and indirect microbe-mediated processes (Li et al., 2013a; Li et al., 2013b; Xue et al., 2016; Tian et al., 2018). In this study, we observed that the increase of soil biological community quantities and diversities enhanced advanced soil nutrients availability. The interactions between cucumber and green garlic roots should be further investigated in future studies.

Effect of intercropping on growth and photosynthetic pigment of cucumber

Continuous mono-cropping of the same crop in the same land can cause a reduction in crop growth and photosynthetic capacity, which was demonstrated in cucumber (Yu et al., 2000; Zhou, Yu & Wu, 2012). Chlorophyll, the main photosynthetic pigment, plays an important role in determining the photosynthetic rate and dry matter production (Dai et al., 2013). Zhou, Cheng & Meng (2007) found that plant biomass and chlorophyll content of the tested tomato and hot pepper was increased by garlic root exudates. In the present study, intercropping with green garlic improved the shoot and root biomass and chlorophyll b content of cucumber, which might be due to stimulation of green garlic root exudates. The same phenomenon was also found in intercropping of garlic with pepper (Ahmad et al., 2013), eggplant (Wang et al., 2015) and tomato (Liu et al., 2014).

Conclusion

The greater shoot biomass of cucumber was observed under the intercropping system than the monocropping system one month after co-existence. Soil analysis confirmed that intercropping with green garlic alleviated the continuous cropping obstacle of cucumber by improving the soil microbial community diversity and composition, as well as soil enzyme activities and nutrient availability. Further studies that focus on underground root interactions and morphogenesis, and persistent effect of green garlic-based intercropping system should be conducted.

Supplemental Information

Data S1 Raw data exported from gas chromatography applied for data analyses and preparation for Table 2 and Figs. 5–7

Click here for additional data file.

Data S2 Raw data for the analyses of biomass, chlorophyll content, soil chemical properties, soil enzymes and soil microbe and preparation for Tables 1 and 3 and Figs. 1–4

Click here for additional data file.

We express our sincere thanks to Muhammad Azam Khan (PMAS-Arid Agriculture University, Rawalpindi, Pakistan) for English language revision and valuable suggestions to the revision of the manuscript.

Additional Information and Declarations

Competing Interests

Author Contributions

Data Availability

The authors declare there are no competing interests.

Xuemei Xiao conceived and designed the experiments, performed the experiments, analyzed the data, prepared figures and/or tables, authored or reviewed drafts of the paper, approved the final draft.

Zhihui Cheng and Jihua Yu conceived and designed the experiments, contributed reagents/materials/analysis tools, authored or reviewed drafts of the paper, approved the final draft.

Jian Lv performed the experiments, analyzed the data, prepared figures and/or tables, approved the final draft.

Jianming Xie analyzed the data, contributed reagents/materials/analysis tools, prepared figures and/or tables, approved the final draft.

Ning Ma performed the experiments, approved the final draft.

The following information was supplied regarding data availability:

The raw data are available in the Supplemental Files.

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
