# Peer review of "A green garlic (Allium sativum L.) based intercropping system reduces the strain of continuous monocropping in cucumber (Cucumis sativus L.) by adjusting the micro-ecological environment of soil"

_PeerJ, doi:10.7717/peerj.7267_

## Round 0.1 · original submission · Major Revisions

Please carefully address the comments of both reviewers.

·

Basic reporting

The manuscript is written in clear English language, easy to read and to understand. The authors explain the prerequisite (the problem) and find a possible explanation, by combined knowledge from literature and from empiric observations in cucumber farming. Experimental design, results and discussion are all dedicated to prove or falsify the hypothesis.

Experimental design

The pot experiment contains the usual elements for verification and falsification of the authors hypothesis. Microbial community characterisation via PLFA reads a bit old-fashioned - since DNA based metagenome analysis methods have become comparably cheap and easily available. The results deriving from PLFA are acceptable, however, the chance to get a more detailed insight in the changes of microbial soil community changes is missed.
There is one specific minor flaw in the methodology, line 95: The addition of mineral nutrients was done in form of an organic fertilizer. It is specified by its nutrient contents, but not called by name or by type.

Validity of the findings

The experiment was done three times with 10 pots of each variant. The results are valuable for both, for research and for cucumber production, although the total effect on cucumber growth was not very high. Organic fertilizers release their nutrients over time, according to the biodegradation progress. A question remains open: can the observed effect on cucumber growth be related to increased nutrient mobilisation from this organic fertiliser?

Additional comments

No more comments than already given in the other sections.

Reviewer 2 ·

Basic reporting

The English writing is hard to understand, and there are too many grammar/tense errors in the manuscript. Please ask a native English speaker for help. Line 81-82 is too colloquial. In Line 153,157 et al. 37°C there should be a space between the number and unit. Please check thoroughly in the whole text.

Experimental design

no comment

Validity of the findings

Please see in General comments.

Additional comments

1. In the introduction part the authors said (Line37-39)the continuous cropping obstacle will cause crop yield and quality reduction, why donot check the cucumber yield and quality?
2. Why choose t-test not ANOVA? Since monocropping and intercropping treatments were compared, an ANOVA analysis is recommended.
3.In all figures please mention n value.
4. In discussion part, you found the differences of soil microbial structure and diversity after intercropping, please add the explaination the relation between this differces and croping obstacle reducing.
5. Less of "Growth and Photosynthetic Pigment Analyses" in discussion part.
6. It seems the authors are very casual to draft the MS. For example,Line63: .... and C and N microbial biomass?????? Line 68:... improved P uptake????
7. In refernce Xiao et al. 2012, 2013, they did garlic-cucumber system in plastic tunnel,in discussion can make a comparasion of this two croping method.

---

## Round 0.2 · accepted · Accept

I have personally checked the revised manuscript file, and read the rebuttal letter. I am satisfied with the changes made by the authors in response to the reviewers’ concerns.